# Can Transfer Entropy Infer Information Flow in Neuronal Circuits for Cognitive Processing?

**DOI:** 10.3390/e22040385

**Published:** 2020-03-28

**Authors:** Ali Tehrani-Saleh, Christoph Adami

**Affiliations:** 1Department of Computer Science and Engineering, Michigan State University, East Lansing, MI 48824, USA; tehrani3@msu.edu; 2BEACON Center for the Study of Evolution, Michigan State University, East Lansing, MI 48824, USA; 3Department of Microbiology & Molecular Genetics, Michigan State University, East Lansing, MI 48824, USA; 4Department of Physics & Astronomy, Michigan State University, East Lansing, MI 48824, USA

**Keywords:** transfer entropy, information flow, neural processing, motion detection, sound localization

## Abstract

How cognitive neural systems process information is largely unknown, in part because of how difficult it is to accurately follow the flow of information from sensors via neurons to actuators. Measuring the flow of information is different from measuring correlations between firing neurons, for which several measures are available, foremost among them the Shannon information, which is an undirected measure. Several information-theoretic notions of “directed information” have been used to successfully detect the flow of information in some systems, in particular in the neuroscience community. However, recent work has shown that directed information measures such as transfer entropy can sometimes inadequately estimate information flow, or even fail to identify manifest directed influences, especially if neurons contribute in a cryptographic manner to influence the effector neuron. Because it is unclear how often such cryptic influences emerge in cognitive systems, the usefulness of transfer entropy measures to reconstruct information flow is unknown. Here, we test how often cryptographic logic emerges in an evolutionary process that generates artificial neural circuits for two fundamental cognitive tasks (motion detection and sound localization). Besides counting the frequency of problematic logic gates, we also test whether transfer entropy applied to an activity time-series recorded from behaving digital brains can infer information flow, compared to a ground-truth model of direct influence constructed from connectivity and circuit logic. Our results suggest that transfer entropy will sometimes fail to infer directed information when it exists, and sometimes suggest a causal connection when there is none. However, the extent of incorrect inference strongly depends on the cognitive task considered. These results emphasize the importance of understanding the fundamental logic processes that contribute to information flow in cognitive processing, and quantifying their relevance in any given nervous system.

## 1. Introduction

When searching for common foundations of cortical computation, more and more emphasis is being placed on information-theoretic descriptions of cognitive processing [1,2,3,4,5]. One of the core tasks in the analysis of cognitive processing is to follow the flow of information within the nervous system, by finding cause-effect components. Indeed, understanding causal relationships is considered to be fundamental to all natural sciences [6]. However, inferring causal relationships and separating them from mere correlations is difficult, and the subject of ongoing research [7,8,9,10,11]. The concept of *Granger causality* is an established statistical measure that aims to determine directed (causal) functional interactions among components or processes of a system. Schreiber [12] described Granger causality in terms of information theory by introducing the concept of *transfer entropy* (TE). The main idea is that if a process *X* is influencing process *Y*, then an observer can predict the future state of *Y* more accurately given the history of both *X* and *Y* (written as $X_{t}^{\left(k\right)}$ and $Y_{t}^{\left(\ell \right)}$, where *k* and *ℓ* determine how many states from the past of *X* and *Y* are taken into account) compared to only knowing the history of *Y*. According to Schreiber, the transfer entropy ${\mathrm{TE}}_{X\to Y}$ quantifies the flow of information from process *X* to *Y*:(1)$$\begin{array}{ccc} {\mathrm{TE}}_{X\to Y} & = & I\left(Y_{t+1}:X_{t}^{\left(k\right)}|\;Y_{t}^{\left(\ell \right)}\right)=H\left(Y_{t+1}|\;Y_{t}^{\left(\ell \right)}\right)-H\left(Y_{t+1}|Y_{t}^{\left(\ell \right)},X_{t}^{\left(k\right)}\right)\\ & = & \sum_{y_{t+1}}\sum_{x_{t}^{\left(k\right)}}\sum_{y_{t}^{\left(\ell \right)}}p\left(y_{t+1},x_{t}^{\left(k\right)},y_{t}^{\left(\ell \right)}\right)\log\frac{p(y_{t+1}|x_{t}^{\left(k\right)},y_{t}^{\left(\ell \right)})}{p(y_{t+1}|y_{t}^{\left(\ell \right)})}\;. \end{array}$$

Here as before, $X_{t}^{\left(k\right)}$ and $Y_{t}^{\left(\ell \right)}$ refer to the history of the processes *X* and *Y*, while $Y_{t+1}$ refers to the variable at t+1 only. Further, $p(y_{t+1},x_{t}^{\left(k\right)},y_{t}^{\left(\ell \right)})$ is the joint probability of $Y_{t+1}$ and the histories $X_{t}^{\left(k\right)}$ and $Y_{t}^{\left(\ell \right)}$, while $p(y_{t+1}|x_{t}^{\left(k\right)},y_{t}^{\left(\ell \right)})$ and $p(y_{t+1}|y_{t}^{\left(\ell \right)})$ are conditional probabilities.

The transfer entropy (Equation 1) is a conditional mutual entropy, and quantifies what the process *Y* at time t+1 knows about the process *X* up to time *t*, given the history of *Y* up to time *t* (see [13] for a thorough introduction to the subject). Specifically, ${\mathrm{TE}}_{X\to Y}$ measures “how much uncertainty about the future course of *Y* can be reduced by the past of *X*, given *Y*’s own past.” Transfer entropy reduces to Granger causality for so-called “auto-regressive processes” [14] (which encompasses most biological dynamics), and has become one of the most widely used directed information measures, especially in neuroscience (see [5,13,15,16] and references cited therein).

While transfer entropy is sometimes used to infer causal influences between susbsystems, it is important to point out that inferring causal relationships is different from inferring information flow [17]. In complex systems (for example, in computations that a brain performs to choose the correct action given a particular sensory experience) events in the sensory past can causally influence decisions significantly distant in time, and to capture such influences using the transfer entropy concept requires a careful analysis in which not only the history lengths *k* and *ℓ* used in Equation (Equation 1) must be optimized, but false influences due to linear mixing of signals (which can mimic causal influences) must also be corrected for [13,15]. In some sense, inferring information flow is a much simpler task than finding all causal influences, as we need only to identify (and quantify) the sources of information transferred to a particular variable. More precisely, for this application the pairwise transfer entropy is used to find candidate sources (in the immediate past) that account for the entropy of a particular neuron.

Using transfer entropy to search for and detect directed information was shown to lead to inaccurate assessments in simple case studies [18,19]. For instance, James et al. [18] presented two examples in which TE underestimates the flow of information from inputs to output in one example, and overestimates it in the other. In the first example, they define a simple system with three binary variables *X*, *Y*, and *Z* where $Z_{t+1}=X_{t}⊕Y_{t}$ (⊕ is the exclusive OR logic operation) and variables *X* and *Y* take states 0 or 1 with equal probabilities, i.e., P(X=0)=P(X=1)=P(Y=0)=P(Y=1)=0.5 (this 2-to-1 relation is schematically shown in Figure 1A). In this network, ${\mathrm{TE}}_{X\to Z}={\mathrm{TE}}_{X\to Z}=0$ whereas the entropy of the process *Z*, H(Z)=1bit, and variables *X* and *Y* certainly influence the future state of *Z*. In this example, the entropy of *Z* can be reduced by 1 bit but the TE does not attribute this entropy to either variables *X* or *Y* and as a consequence the TE underestimates the flow of information from *X* and *Y* to *Z*. In another example, they define a system with two binary variables *Y* and *Z*, where $Z_{t+1}=Y_{t}⊕Z_{t}$ and similar to the previous example, P(Y=0)=P(Y=1)=P(Z=0)=P(Z=1)=0.5 (this feedback loop relation is schematically shown in Figure 1B). In this scenario, ${\mathrm{TE}}_{Y\to Z}=1\;\mathrm{bit}$, which implies that the entire 1 bit of entropy in *Z* is coming from process *Y*. However, this is not correct since both *Y* and *Z* are equally contributing to determine the future state of *Z*. In this example, TE overestimates the information flow from process *Y* to *Z*. It is also noteworthy that in this example the processed information (defined as $I(Z_{t}:Z_{t+1})$) vanishes, which again does not correctly detect the other source, $Z_{t}$, from which the information is coming. As acknowledged by the authors in [18], expecting that the entropy of the output $H(Z_{t+1})$ is given simply by the sum of the transfer entropy from each of the inputs independently is a naive interpretation of information flow. Indeed, this is generally not the case, even if the two sources are uncorrelated. Consider for example, the first system described above in which $Z_{t+1}=f\left(X_{t},Y_{t}\right)$. Suppose *f* is a deterministic function of $X_{t}$ and $Y_{t}$, in which case the conditional entropy $H(Z_{t+1}|X_{t},Y_{t})=0$. Then, the entropy $H(Z_{t+1})$ decomposes into the sum of an unconditional and a conditional transfer entropy
(2)$$\begin{array}{c} H\left(Z_{t+1}\right)\;=\;{\mathrm{TE}}_{Y\to Z}+{\mathrm{TE}}_{X\to Z|Y_{t}}\;, \end{array}$$
where the conditional transfer entropy is defined as (see [13], Section 4.2.3)
(3)$$\begin{array}{c} {\mathrm{TE}}_{Y\to Z|X_{t}}=I\left(Y_{t}:Z_{t+1}|Z_{t},X_{t}\right)\;. \end{array}$$

Using this definition, it is easy to show that
(4)$$\begin{array}{c} {\mathrm{TE}}_{Y\to Z}={\mathrm{TE}}_{Y\to Z|X_{t}}+I\left(X_{t}:Y_{t}:Z_{t+1}|Z_{t}\right)\;, \end{array}$$
and Equation (Equation 2) can be rewritten in terms of transfer entropies only, or else conditional transfer entropies only, as
(5)$$\begin{array}{c} H\left(Z_{t+1}\right)\;=\;{\mathrm{TE}}_{Y\to Z|X_{t}}+{\mathrm{TE}}_{X\to Z|Y_{t}}+I\left(X_{t}\;:\;Y_{t}\;:\;Z_{t+1}|Z_{t}\right)\;=\;{\mathrm{TE}}_{Y\to Z}+{\mathrm{TE}}_{X\to Z}-I\left(X_{t}\;:\;Y_{t}\;:\;Z_{t+1}|Z_{t}\right)\;. \end{array}$$

In light of Equation (Equation 5), it then becomes clear that the naive sum of the transfer entropies ${\mathrm{TE}}_{X\to Z}$ and ${\mathrm{TE}}_{Y\to Z}$ (or naive sum of conditional transfer entropies) must fail to account for the entropy of *Z* whenever the term $I(X_{t}\;:\;Y_{t}\;:\;Z_{t+1}|Z_{t})$ is non-zero, and therefore will fail to fully and accurately quantify information transferred from sources *X* and *Y*. Therefore, the error in information flow estimate when using transfer entropy is simply given by the absolute value of $I(X_{t}\;:\;Y_{t}\;:\;Z_{t+1}|Z_{t})$ (same when using conditional transfer entropies).

Now consider the second example system with a feedback loop in which $Z_{t+1}=f\left(Y_{t},Z_{t}\right)$, and again suppose *f* is a deterministic function which implies $H(Z_{t+1}|Y_{t},Z_{t})=0$. In this case, there is a similar information decomposition that now involves a shared entropy $I(Y_{t}\;:\;Z_{t}\;:\;Z_{t+1})$
(6)$$\begin{array}{c} I\left(Y_{t}\;:\;Z_{t+1}\right)={\mathrm{TE}}_{Y\to Z}+I\left(Y_{t}\;:\;Z_{t}\;:\;Z_{t+1}\right)\;. \end{array}$$

Here, the entropy $H(Z_{t+1})$ can be written in terms of transfer entropy and processed information (recall that $H(Z_{t+1}|Z_{t},Y_{t})=0$)
(7)$$\begin{array}{c} H\left(Z_{t+1}\right)={\mathrm{TE}}_{Y\to Z}+I\left(Z_{t}\;:\;Z_{t+1}\right)\;. \end{array}$$

While Equation (Equation 7) shows that the sum of transfer entropy ${\mathrm{TE}}_{Y\to Z}$ and processed information $I(Z_{t}\;:\;Z_{t+1})$ account for all the entropy $Z_{t+1}$, these two terms do not always individually identify the sources of information flow correctly. For instance, we have seen that in the second example (where $Z_{t+1}=Y_{t}⊕Z_{t}$) the processed information $I(Z_{t}\;:\;Z_{t+1})$ vanishes even though variable $Z_{t}$ most definitely influences the state of variable $Z_{t+1}$. As discussed earlier, all the information transferred to $Z_{t+1}$ in that case is attributed to variable $Y_{t}$. Note that the processed information can be written as
(8)$$\begin{array}{c} I\left(Z_{t}\;:\;Z_{t+1}\right)=I\left(Z_{t}\;:\;Z_{t+1}|Y_{t}\right)+I\left(Z_{t+1}\;:\;Z_{t}\;:\;Y_{t}\right) \end{array}$$
where $I(Z_{t}\;:\;Z_{t+1}|Y_{t})=1$ and $I(Z_{t+1}\;:\;Z_{t}\;:\;Y_{t})=-1$.

Note that for the most general case where function *f* can be non-deterministic and the network with or without feedback loop, the full entropy decomposition can be written as
(9)$$\begin{array}{c} H\left(Z_{t+1}\right)={\mathrm{TE}}_{Y\to Z|X_{t}}+{\mathrm{TE}}_{X\to Z|Y_{t}}+I\left(X_{t}\;:\;Y_{t}\;:\;Z_{t+1}|Z_{t}\right)+I\left(Z_{t+1}\;:\;Z_{t}\right)+H\left(Z_{t+1}|X_{t},Y_{t},Z_{t}\right)\;. \end{array}$$

There is also another key factor in the examples described above that results in misestimating information flow when using transfer entropy. In both examples, the input to output relation is implemented by an XOR function. For instance, in the first example ($Z_{t+1}=X_{t}⊕Y_{t}$), the transfer entropy ${\mathrm{TE}}_{X\to Z}$ considers *X* in isolation and independent of variable *Y*. We should make it clear that it is not the formulation of TE that is at the origin of mis-attributing the sources of the transferred information. Rather, by definition Shannon’s mutual information, I(X:Y)=H(X)+H(Y)−H(X,Y) is dyadic, and cannot capture polyadic correlations where more than one variable influences another. Consider for example a similar but *time-independent* process between binary variables *X*, *Y*, and *Z* where Z=X⊕Y. As is well-known, the mutual information between *X* and *Z*, and also between *Y* and *Z* vanishes: I(X:Z)=I(Y:Z)=0 (this corresponds to the one-time pad, or Vernam cipher [20], a common method of encryption that takes advantage of the fact that I(X:Y:Z)=−1). Thus, while the TE formulation aims to capture a directed dependency of information, Shannon information measures the *undirected* (correlational) dependency of two variables only. As a consequence, problems with TE measurements in detecting directed dependencies are unavoidable when using Shannon information, and do not stem from the formulation of transfer entropy [12] or similar measures such *causation entropy* [10] to capture causal relations. Note that methods such as *partial information decomposition* have been proposed to take into account the synergistic influence of a set of variables on the others [21]. However, such higher-order calculations are more costly (possibly exponentially so) and require significantly more data in order to perform accurate measurements.

Given the observed error in measuring information flow using TE due to logic gates that encrypt, we now set out to examine how well TE measurements capture information flow when the function is implemented with Boolean functions other than XOR. In particular, we examine every first-order Markov process $Z_{t+1}=f\left(X_{t},Y_{t}\right)$ where function *f* is implemented by all 16 possible 2-to-1 binary relations (Figure 1A) and quantify the error in information transfer estimate for each of them. Similar to previous examples, the state of variable *Z* is independent of its past, and inputs *X* and *Y* take states 0 and 1 with equal probabilities, i.e., P(X=0)=P(X=1)=P(Y=0)=P(Y=1)=0.5.

Table 1 shows the results of transfer entropy measurements for all possible 2-to-1 logic gates and the error that would occur if TE measures are used to quantify the information flow from inputs to outputs. This error is the sum of misestimations in information flow quantified by pairwise transfer entropies ${\mathrm{TE}}_{X\to Z}$ and ${\mathrm{TE}}_{Y\to Z}$. As we discussed before, for the XOR relation the transfer entropies ${\mathrm{TE}}_{X\to Z}={\mathrm{TE}}_{Y\to Z}=0$, and $H(Z_{t+1})=1$ which means that TE misestimates the information flow from inputs *X* and *Y* by 1 bit (the XNOR is exactly the same). We find that in all other polyadic relations where both *X* and *Y* influence the future state of *Z*, ${\mathrm{TE}}_{X\to Z}$ and ${\mathrm{TE}}_{Y\to Z}$ capture part of the information flow from inputs to outputs, but ${\mathrm{TE}}_{X\to Z}+{\mathrm{TE}}_{Y\to Z}$ is less than the entropy of the output *Z* by 0.19 bits (${\mathrm{TE}}_{X\to Z}+{\mathrm{TE}}_{Y\to Z}=0.62$, H(Z)=0.81). In the remaining six relations where only one of the inputs or neither of them influences the output, the transfer entropies correctly capture the information flow. The difference between the sum of transfer entropies, ${\mathrm{TE}}_{X\to Z}+{\mathrm{TE}}_{Y\to Z}$, and the entropy of the output H(Z) in XOR and XNOR relations, stems from the fact that I(X:Y:Z)=−1, the tell-tale sign of encryption. Furthermore, while other polyadic gates do not implement perfect encryption, they still encrypt partially I(X:Y:Z)=−0.19, which we call *obfuscation*. It is this obfuscation that is at the heart of the TE error shown in Table 1.

We repeated similar calculations for the case of a feedback loop network where $Z_{t+1}=f\left(Y_{t},Z_{t}\right)$ (Figure 1B) and function *f* can be any one of the 16 logic relations shown in Table 1. These simple calculations show that in 16 relations including XOR and XNOR, the sum of the transfer entropies, ${\mathrm{TE}}_{Y\to Z}+I\left(Z_{t+1}:Z_{t}\right)$ (the formulation for transfer entropy of a variable to itself reduces to processed information $I(Z_{t+1}:Z_{t})$) is equal to the entropy of the output $Z_{t+1}$ as was shown in Equation (Equation 7). However, in XOR and XNOR relations transfer entropy incorrectly attributes all the information to one of the input variables and no influence is attributed to the other. Furthermore, in the polyadic relations other than XOR and XNOR, the transfer entropies ${\mathrm{TE}}_{Y\to Z}$ and $I(Z_{t+1}:Z_{t})$ differ in value while variables *X* and *Y* equally influence the state of the output *Z*, which is why the TE error in these relations is 0.19 bits.

Given that pairwise TE measurements (not taking into account higher-order conditional transfer entropies) only fail to correctly identify the sources of information flow in cryptographic gates and demonstrate partial errors in quantifying information flow in polyadic relations, we now set out to determine how often these relations appear in networks that implement basic cognitive tasks, and how much error is introduced when measuring information flow using transfer entropy. If the total error in transfer entropy measurements of information flow in cognitive networks is significant, an analysis of pairwise directed information among neural components (neurons, voxels, cortical columns, etc.) using this concept is bound to be problematical. If, however, these errors are reasonably low within biological control structures because cryptographic logic is rarely used, then treatments using the TE concept can largely be trusted.

To answer this question, we use a new tool in computational cognitive neuroscience, namely computational models of cognitive processing that can explain task-performance in terms of plausible dynamic components [22]. In particular, we use Darwinian evolution to evolve artificial digital brains (also known as Markov Brains or MBs [23]) that can receive sensory stimuli from the environment, process this information, and take actions in response (In the following we refer to digital brains as “Brains”, while biological brains remain “brains”.). We evolve Markov Brains that perform two different cognitive tasks whose circuitry is thoroughly studied in neuroscience: visual motion detection [24], as well as sound localization [25,26]. Markov Brains have been shown to be a powerful platform that can unravel the information-theoretic correlates of fitness and network structure in neural networks [27,28,29,30,31]. This computational platform enables us to analyze structure, function, and circuitry of hundreds of evolved digital Brains. As a result, we can obtain statistics on the frequency of different types of relations in evolved circuits (as opposed to studying only a single evolutionary outcome), and further assess how crucial different operators are for each evolved task, by performing knockout experiments in order to measure an operator’s contribution to the task. In particular, we first investigate the composition of different types of logic gates in networks evolved for the two cognitive tasks, and then theoretically estimate how accurate transfer entropy measures could be when applied to quantify the pairwise information flow from one neuron to another in such simple cognitive networks. We then use transfer entropy measures as a statistic to identify information flow between neurons of evolved circuits using the time series of neural recordings obtained from behaving Brains engaged in their task, and evaluate how successful transfer entropy is in detecting this flow. While artificial evolution of control structures (“artificial Brains”) is not a substitute for the analysis of information flow in biological brains, this investigation should provide some insights on how accurate (or inaccurate) transfer entropy measures could be.

## 2. Materials and Methods

### 2.1. Markov Brains

Markov Brains (MB) are evolvable networks of binary neurons (they take value 0 for a quiescent neuron, or 1 for a firing neuron) in which neurons are connected via probabilistic or deterministic logic gates (in this work, we constrain MBs to only use 2-to-1 deterministic logic gates). The states of the neurons are updated in a first order Markov process, i.e., the probability distribution of states of the neurons at time step t+1 depends only on the states of neurons at time step *t*. This does not imply that Markov Brains are memoryless, because the state of one neuron can be stored by repeatedly writing into its own (or another) neuron’s state variable [23,27,30]. The connectivity and the underlying logic of the MB’s neuronal network is encoded in a genome. Thus, we can evolve populations of MBs using a Genetic Algorithm (GA) [32] to perform a variety of cognitive tasks (for a more detailed description of Markov Brain function and implementation see [23]). In the following sections, we describe two fitness functions designed to evolve motion detection and sound localization circuits in MBs.

### 2.2. Motion Detection

The first fitness function is designed in order to evolve MBs that function as a visual motion detection circuit. Reichardt and Hassenstein proposed a circuit model of motion detection that is based on a delay-and-compare scheme [33]. The main idea behind this model is that a moving object is sensed by two adjacent receptors on the retina, at two different time points. Figure 2 shows the schematic of a Reichardt detector in which the τ components delay the stimulus and × components multiply the signals, i.e., fires if the signal from the receptor and delay component arrive at the same time. The result of the multiplication units for two different directions is then subtracted so that high values denote motion in one direction (the “preferred direction”, PD), low values denote the opposite direction (null direction, ND), and intermediate values encode a stationary stimulus.

The experimental setup for evolution of motions detection circuits is similar to the setup previously used in [34]. In that setup, two sets of inputs are presented to a MB at two consecutive times and the Brain classifies the input as preferred direction (PD), stationary, or null direction (ND). After the first set of inputs, i.e., at time *t* in Figure 2B, a Markov Brain is updated once, and after the second set of inputs, at t+1, it is updated two times which simulates two operations performed after delaying one of the inputs, namely multiplication and subtraction. The value of the sensory neuron becomes 1 when a stimulus is present, and it becomes 0 otherwise (see Figure 2B). Thus, 16 possible sensory patterns can be presented to the MB to classify, among which 3 input patterns are PD, 3 are ND, and the other 10 are stationary patterns. Two neurons are assigned as output neurons of the motion detection circuit. The sum of binary values of these neurons represents the output of the motion detection circuit, 0: ND, 1: stationary stimulus, 2: PD, while in the Reichardt detector circuit shown in Figure 2A, the output corresponding to ND is −1, stationary is 0, and PD is +1.

### 2.3. Sound Localization

The second fitness function is designed to evolve MBs that function as a sound localization circuit. Sound localization mechanisms in mammalian auditory systems function based on several cues such as interaural time difference, interaural level difference, etc. [35]. Interaural time difference (which is the main cue behind the sound localization mechanism) is the difference between the times at which sound reaches the two ears. Figure 3A shows a simple schematic of a sound localization model proposed by Jeffress [36] in which sound reaches the right ear and left ear at two possibly different times. These stimuli are then delayed in an array of *delay* components and travel to an array of detector neurons (marked with different colors in Figure 3A). Each detector only fires if the two signals from different pathways, the left ear pathway (shown in bottom) and the right ear pathway (shown in top), reach that neuron simultaneously.

In our experimental setup, two sequences of stimuli are presented to two different sensory neurons (neurons $N_{0}$ and $N_{1}$) that represent the receptors in the two ears. The stimulus in two sequences are lagged or advanced with respect to one another (as shown in Figure 3B). The agent receives these sequences and should identify 5 different angles from where that sound is coming. The binary value of the sensory neuron becomes 1 when a stimulus is present, shown as black blocks in Figure 3B, and it becomes 0 otherwise, shown as white blocks in Figure 3B. Markov Brains are updated once after each time step in the experiment. Similar to schema shown in Figure 3A, Markov Brains have five designated output neurons ($N_{11}$–$N_{15}$) and each neuron corresponds to one of the sound sources placed at a specific angle. Colors of detector neurons ($N_{11}$–$N_{15}$) in Figure 3B match the angle of each sound source in Figure 3A.

## 3. Results

For the motion detection (MD) and sound localization (SL) tasks, we evolved 100 populations each for 10,000 generations, allowing all possible 2-to-1 (deterministic) logic gates as primitives. At the end of each evolutionary run, we isolated one of the genotypes with the highest score from each population to generate a representative circuit.

### 3.1. Gate Composition of Evolved Circuits

Out of 100 populations evolved in motion detection task, 98 led to circuits that perform motion detection with perfect fitness. The number of gates in evolved Brains varies tremendously, with a minimum of four and maximum of 17 (mean = 7.92, SD = 2.48). The frequency distribution of types of logic gates per each individual Brain is shown for these 98 perfect circuits in Figure 4A (in this figure, AND-NOT is an asymmetric AND operation where one of the variables is negated, for example $X^{\prime }\cdot Y$. Similarly, OR-NOT is an asymmetric OR operation, e.g., $X+Y^{\prime }$). To gain a better understanding of the distribution of logic gates and how they compose the evolved motion detection circuits, we performed gate-knockout assays on all 98 Brains. We sequentially eliminated each logic gate, where we also eliminate all the input and output connections of that gate, and re-measured the mutant Brain’s fitness, thus allowing us to estimate which gates were essential to the motion detection function (if there is a drop in mutant Brain’s fitness) and which gates were redundant to the motion detection function (if a mutant Brain’s fitness remains perfect). The frequency distribution of each type of logic gate per individual Brain for essential gates is shown for the 98 perfect Brains in Figure 4B.

For the sound localization task, 71 evolution experiments out of 100 resulted in Markov Brains with perfect fitness. The minimum number of gates was six, with a maximum of 15 (mean = 9.14, SD = 1.77). Figure 4A shows the frequency distribution of types of logic gates per Brain for these 71 perfect Brains. We also performed a knockout analysis on all 71 evolved sound localization Brains. The frequency distribution of each type of logic gate per individual Brain for essential gates is shown for the 71 perfect Brains in Figure 4B. These results demonstrate that the gate type compositions and circuit structures in evolved Brains for motion detection (MD) and sound localization (SL) tasks are significantly different. The total number of logic gates (ignoring duplicates) in the SL task (9.14 gates per Brain, SD = 1.77) is greater than the total number of gates in the MD task (7.92 gates per Brain, SD = 2.48). Moreover, the number of essential gates in SL (7.13 gates per Brain, SD = 1.24) is also greater than the number of essential gates in MD (5.23 gates per Brain, SD = 1.31).

### 3.2. Transfer Entropy Misestimates Caused by Encryption or Polyadicity

As discussed earlier, transfer entropy measures may misestimate the information flow from input to output and may fail to correctly identify the source of information. Table 1 gave a detailed analysis of transfer entropy measurements and their misestimates that are rooted either in the polyadic or encrypting nature of the gate, for all possible 2-to-1 logic gates. Given the gate distributions of evolved circuits for motion detection and sound localization tasks along with the misestimate values calculated in Table 1, we can estimate the error that would occur when using transfer entropy to quantify the pairwise information flow from source neurons (i.e., input neurons to gates) to receiver neurons (i.e., output neurons of gates). We can similarly estimate what fraction of the information flow from inputs to outputs would be *correctly* quantified by the transfer entropy in the evolved circuits. Recall that in the results presented in Table 1, calculations were performed assuming that the input bits take values 0 or 1 with equal probability 0.5. Of course, we cannot generally assume this for the input bits of every logic gate in an evolved network. As a consequence, this analysis only approximates the information flow misestimates of the full network.

In our analysis, we only evaluated the contribution of gates deemed essential via the knockout test. For these essential gates, we summed the pairwise information flow misestimates as well as the correct information flow attributions in each evolved Brain. The mean values of calculated misestimates of information flow as well as correct measurements with their 95% confidence intervals for 98 evolved circuits that perform motion detection task, and for 71 evolved sound localization Brains are shown in Figure 5A. In Figure 5B, we normalized misestimates and correct measurements by dividing by the number of essential gates in each Brain, and averaged them across Brains. It is worth noting that the calculated information flow misestimates shown in these plots only reflect the misestimates that originated from the polyadicity or encrypting nature of the gates, since they are only based on the network structure and the gate composition of each Brain as well as the analytical results presented in Table 1, and do not take into account the errors that could occur as a result of factors such as sampling errors in the dataset or structural complexities in the network, such as recurrent or transitive relations [10,11,37]. Along the same line of reasoning, calculated values of correct measurements represent correct information flows that could be measured by transfer entropy in the absence of the aforementioned sources of errors.

These results further reveal that the circuit structures and gate type compositions in the two tasks are significantly different, and that this structural difference leads to different outcomes when transfer entropy measures are used to detect pairwise information flows. Transfer entropy can potentially capture 3.31 bits (SE = 0.10) of information flow correctly in evolved motion detection circuits (0.64 bits per gate, SE = 0.014), and 3.95 (averaged across 71 Brains, SE = 0.14) bits in evolved sound localization circuits (0.55 bits per gate, SE = 0.014). However, the information flow misestimates when using transfer entropy in evolved sound localization circuits is 2.39 bits (averaged across 71 Brains, SE = 0.12) which is significantly higher than the misestimates in evolved motion detection circuits, which is 1.33 bit (average across 98 Brains, SE = 0.085). The information flow misestimate in evolved motion detection circuits is 0.25 bits per gate (SE = 0.014) whereas it is 0.34 bits (SE = 0.016) per gate in evolved sound localization circuits. These findings show that the accuracy of transfer entropy measurements for detecting information flow in digital neural networks can vary significantly from one task to another.

### 3.3. Transfer Entropy Measurements from Recordings of Evolved Brains

In the previous section we estimated errors in information flow attribution using the error that each particular logic gate in Table 1 entails, and then calculating the total error using the gate type distribution for each cognitive task. However as mentioned earlier, this approach only gives a crude estimate of flow because in the evolved cognitive circuits the neurons (and therefore the logic gates) are not independent, and their input is not in general maximum entropy.

Here we use a different approach to assess transfer entropy measurement accuracy in identifying inter-neuronal relations of evolved Markov Brains: we record the neural activities of an evolved Brain when performing a particular cognitive task, similar to the neural recording (“brain mapping”) performed on behaving animals. We collect the recordings in all possible trials for each cognitive task and create a dataset for each evolved Brain for that cognitive task. More precisely, for Brains that evolved to perform the motion detection task we record neural firing patterns in 16 different trials. At the beginning of each trial, the Brain is in a state in which all neurons are quiescent. Then, the Brain is updated three times, so we record the Brain’s neural activity in 4 consecutive time steps (including the initial state). As a result, the recordings dataset of a Brain that performs motion detection consists of 64 snapshots of the Brain, i.e., the binary state of each neuron. Similarly, a Brain that performs sound localization is recorded during five different trials and during each trial the Brain is recorded in four consecutive time steps. This results in a recording dataset of size 20 for each evolved Brain. Note that these evolved Brains are deterministic, thus, if a Brain is recorded in the same trial multiple times, its behavior and neural activities remain exactly the same and therefore, recording a Brain once in each trial is sufficient. We then use these recordings to measure transfer entropy for every pair of neurons ${\mathrm{TE}}_{N_{i}\to N_{j}}$ in the network. These transfer entropy measures can be used as a statistic to test whether a neuron $N_{i}$ causally influences another neuron $N_{j}$. Figure 6A shows the result of TE calculations performed on neural recording for a Markov Brain evolved in the sound localization task.

To test the accuracy of the TE prediction, we construct an influence map for each neuron of the Markov Brain that shows which other neurons are influenced by a particular neuron. Such a mapping also determines the receptive field of each neuron, which specifies which other neurons influence a particular neuron. Markov Brains evolve complex networks in which multiple logic gates can write to the same neuron and as a result, it is not straightforward to deduce input-output relations among neurons. Indeed, it was previously argued that even with a complete knowledge of a given system, finding the causal relation among the components of the system may be a very difficult task [8,38,39].

To create our “ground truth” model of direct influence relations, we take into consideration two different components of a Brain’s network. First, we take into account the input neurons of a gate and its output neuron, while we also take into consideration the type of the logic gate. For example, in the case of a ’ZERO’ gate where the output is always 0 we do not interpret this connection to reflect information flow (as there is no entropy in the output). Second, we analytically extract the binary state of each neuron as a Boolean function of all other neurons using a logic table of the entire Brain (logic table of size $2^{16}$, for 16 neurons). This helps us rule out neurons that are connected as inputs to a logic gate while not actually contributing to the output neuron of that gate. Note that this procedure is specifically helpful in cases where more than one logic gate writes into a neuron (when more than one gate writes into a neuron the ultimate result is the bitwise OR of all incoming signals since if either one of them is a non-zero signal it would make the neuron fire, i.e., its state becomes 1). Figure 6B shows an example of “ground truth” influence map of neurons for a Brain evolved for sound localization. Each row of this plot shows the influence map of the corresponding neuron and each column represents the receptive field of that neuron. Note that in this plot values are binary, i.e., they are either 0 or 1 which specifies whether a source neuron influences a destination neuron, whereas TE measurements vary in the range [0, 1] bits. Keep in mind that this influence map is only an estimate of information flow gathered from gate logic and connectivity shown in Figure 6C.

In order to compare TE measurements with influence maps, we first assume that any non-zero value of the ${\mathrm{TE}}_{N_{i}\to N_{j}}$ implies that there is some flow of information from neuron $N_{i}$ to $N_{j}$. We then evaluate how well TE measurements detect the information flow among neurons based on this assumption. In particular, for each evolved Brain we count 1) the number of existing pairwise information flows between neurons that is correctly detected by TE (hit), 2) the number of relations that are present in the influence map but were not detected by TE (miss), and 3) the number of existing pairwise information flow between the neurons detected by TE measurements that according to the influence map were incorrectly detected (false-alarm). Figure 7A,B show the performance results of TE measurements in detecting information flow in Brains evolved in motion detection and sound localization, respectively (averaged across best performing Brains and 95% confidence interval). We observe that the number of false-alarms in motion detection (mean = 19.0, SE = 0.86) is greater the number of hits (mean = 6.8, SE = 0.20). Similarly, in sound localization the number of false-alarms (mean = 45.1, SE = 1.63) is also greater than the number of hits (mean = 10.1, SE = 0.31), but significantly more so. This again underscores that the accuracy of transfer entropy measures strongly depends on the characteristics of the task that is being solved.

In the results shown in Figure 7A,B, we assumed that any value of transfer entropy greater than 0 implies information flow. This assumption can be relaxed such that only transfer entropy values that are greater than a particular threshold imply information flow. We calculated TE measurement performance for a variety of threshold values in the range [0,1]. The results are presented as receiver operating characteristic (ROC) curves that show hit rates as a function of false-alarm rates as well as their 95% confidence intervals in Figure 7C,D for motion detection and sound localization, respectively [40]. In these plots, the dashed line shows a fitted ROC curve assuming a Gaussian distribution for the p(TE|informationflowispresent) and p(TE|informationflowisnotpresent). The resulting ROC function is $f\left(x\right)=\frac{1}{2}erfc\left(\frac{\mu_{1}-\mu_{2}}{\sqrt{2}\sigma_{2}}+\frac{\sigma_{1}}{\sigma_{2}}erfc^{-1}\left(2x\right)\right)$, where erfc is the “error function” complement and $erfc^{-1}$ is the inverse of the error function complement.

In the ROC plots, the datapoint with the highest hit rate (right-most data point) is the normalized result shown in Figure 7A,B, that is, the analysis with a vanishing threshold. Note also that the data in Figure 7 represent hit rates against false-alarm rate for thresholds spanning the entire range [0,1], implying that hit rates cannot be increased any further unless we assume there is an information flow between every pair of neurons (hit rate = false-alarm rate = 1). The false-alarm rates in the ROC curves are actually fairly low in spite of the significant number of false alarms we see in Figure 7A,B. This is due to the fact that the number of existing pairwise information flows in a Brain network is much smaller than the number of non-existing flows between any pair of neurons (the influence map matrices are sparse). Thus, when dividing the number of false-alarms by the total number of non-existing information flows, the false-alarm rate is low.

## 4. Discussion

We used an agent-based evolutionary platform to create digital Brains so as to quantitatively evaluate the accuracy of transfer entropy measurements as a proxy for measuring information flow. To this end, we measured the frequency and significance of cryptographic and polyadic 2-to-1 logic gates in evolved digital Brains that perform two fundamental and well-studied cognitive tasks: visual motion detection and sound localization. We evolved 100 populations for each of the cognitive tasks and analyzed the Brain with the highest fitness at the end of each run. Markov Brains evolved a variety of neural architectures that vary in number of neurons and the number of logic gates, as well as the type of logic gates to perform each of the cognitive tasks. In fact, both modeling [41] and empirical [42] studies have shown that a wide variety of internal parameters in neural circuits can result in the same functionality [43]. Thus, it would be informative and perhaps necessary to examine a variety of circuits that perform the same cognitive task [34].

An analysis of the evolved Brains suggests that selecting for different cognitive tasks leads to significantly different gate type distributions. Using the error estimate for each particular gate due to encryption or polyadicity, we used the gate type distributions for each cognitive task to estimate the total error in information flow stemming from using transfer entropy as a statistic. The transfer entropy misestimate was 1.33 bits (SE = 0.08) per Brain on average for Brains evolved for motion detection, whereas in evolved Brains performing sound localization the misestimate was significantly higher: 2.39 bits (SE = 0.12) per Brain on average. More importantly, the inherent differences between the two tasks result in different levels of accuracy when using transfer entropy measures to identify information flow between neurons. It is important to note that in calculating these misestimates, we only accounted for the misestimates that result from TE measurements in polyadic or cryptographic gates. However, we commonly face several other challenges when applying the transfer entropy concept to components of nervous systems (neurons, voxels, etc.). These challenges range from intrinsic noise in neurons to inaccessibility of recording data for larger populations of neurons which we discuss in more detail later.

We also tested how well transfer entropy can identify the existence of information flow between any pair of neurons using the statistics of neural recordings at two subsequent time points only. Because a perfect model for the “ground truth” of information flow is difficult (if not impossible) to establish, we use an approximate ground truth that uses the connectivity of the network, along with information from the (simplified) logic function to provide a comparison. We find that TE captures many of the connections established by the ground truth model, with a true positive rate (hit rate) of 73.1% for motion detection and 78.7% for sound localization (assuming any non-zero value of transfer entropy implies causal relation). The TE measurements miss some relations from the established ground truth while also providing demonstrably false positives, with a false-alarm rate of 7.7% in motion detection and 18.5% for sound localization. For example, some of the information flow estimates in Figure 6 manifestly reverse the actual information flow, suggesting a backwards flow that is causally impossible. Such erroneous backwards influence is possible, for example, when the signal has a periodicity that creates accidental correlations with significant frequency. Besides these false positives, the false negatives (missed inferences) are due to the use of information-hiding (cryptographic or obfuscating) relations, as discussed earlier.

It is noteworthy that in the transfer entropy measurements we performed, we benefited from multiple factors that are commonly great challenges in TE analysis of biological neural recordings. First, our TE measurement results were obtained using error-free recordings of noise-free neurons, while biological neurons are intrinsically noisy. We were also able to use the recordings from every neuron in the network, which presumably results in more accurate estimates. In contrast, in biological networks we only have the capacity to record from a finite number of neurons which, in turn, constrains our understanding of how information flows in the network.

Furthermore, by focusing only on information flow from one time step to the next we can evade the complex issues posed by estimating causal influence, which requires finding optimal time delays in transfer entropies. For example, while a signal may influence a neuron’s firing three time steps after it was perceived by a sensory neuron, it must be possible to follow this influence step-by-step in a first-order Markov process, as causal signals must be relayed physically (no action-at-a-distance). As a consequence, when using transfer entropy to detect and follow information flow, we can restrict ourselves to history lengths of 1 (k=l=1), which significantly simplifies the analysis [17]. Furthermore, complications arising from discretizing continuous signals [15] do not arise, nor is there a choice in embedding the signal as all our neurons have discrete states. In principle, extending the history lengths (from k=ℓ=1 to higher) may be used to get rid of false positives in entropy estimates (even for a first-order Markov process), for the simple reason that the higher dimensionality of state space reduces accidental correlations, given a finite sample set. However, such an increase in dimensionality has a drawback: it makes the detection of true positives more difficult (it increases the rate of false negatives) unless the dataset size is also increased. In many dynamical systems such an increase in data size is not an issue, but it may be very difficult (if not impossible) for smaller systems such as the simple cognitive circuits that we evolve. For those, the number of different “sensory experiences” is extremely limited, and increasing the dataset size does not solve the problem because it would simply repeat the same data. In other words, unlike for large probabilistic systems where generating longer time series will almost invariably exhaustively sample the probability space, this is not the case for motion detection and sound localization. For such “small” systems, increasing the history lengths reduces false positives, but increases false negatives at the same time.

Finally, in order to precisely calculate transfer entropy from Equation (Equation 1), the summation should be performed over all possible states of variables $X_{t}$, $Y_{t}$, $Y_{t+1}$. Using only a subset of those states when calculating the entropy estimate may result in false positives, as well as false negatives. This is another common source of inaccuracy in TE measurements of neural recordings. Here we were able to generate neural recording data for all possible sensory input patterns and included them in our dataset, yet we still observe the described shortcomings in our results. This brings up another important point to notice, namely, even if we introduce every possible sensory pattern to the network, we do not necessarily observe every possible neural firing pattern in the network, and as a result, we do not necessarily sample the entire set of variable states $\left(Y_{t+1},Y_{t},X_{t}\right)$.

## 5. Conclusions

Our results imply that using pairwise transfer entropy has its limitations in accurately estimating the information flow, and its accuracy may depend on the type of network or cognitive task it is applied to, as well as the type of data that is used to construct the measure. Higher-order conditional transfer entropies or more sophisticated measures such as partial information decomposition [21] may be able to alleviate those errors, at the expense of significant computational investments. We also find that simple networks that respond to a low-dimensional set of stimuli (such as the two example tasks investigated here) lead to problems in inferring information flow simply because transfer entropy estimates will be prone to sampling errors.

These findings highlight the importance of understanding the frequency and types of fundamental processes and relations in biological nervous systems. For example, one approach would be to examine transfer entropy in known systems, especially in simple biological neural networks in order to shed light on the strengths and deficiencies of current methods. Performing an information flow analysis on brains in vivo will remain a daunting task for the foreseeable future, but advances in the evolution of digital cognitive systems may allow us a glimpse of the circuits in biological brains, and perhaps guide the development of other measures of information flow.

## Figures and Tables

**Figure 1 entropy-22-00385-f001:**
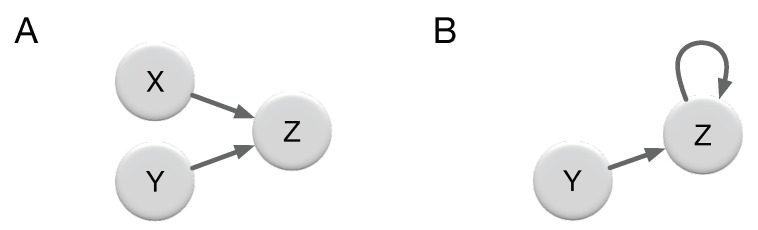
(**A**) A network where processes *X* and *Y* influence future state of *Z*, $Z_{t+1}=f\left(X_{t},Y_{t}\right)$. (**B**) A feedback network in which processes *Y* and *Z* influence future state *Z*, $Z_{t+1}=f\left(Y_{t},Z_{t}\right)$.

**Figure 2 entropy-22-00385-f002:**
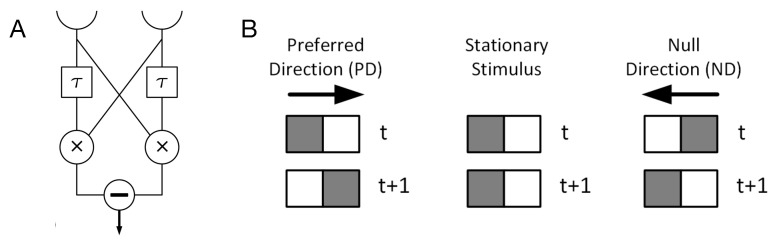
(**A**) A Reichardt detector circuit. In this circuit, the results of the multiplications from each pathway are subtracted to generate the response. The circuit’s outcome for PD is +1, ND is −1, and for stationary patterns is 0. (**B**) Schematic examples of three types of input patterns received by the two sensory neurons at two consecutive time steps. Grey squares show presence of the stimuli in those neurons. The sensory pattern shown here for PD is 10 at time *t* and 01 at time t+1, which we write as: 10→01. Patterns 11→01 and 00→10 also represent PD. Similarly, pattern 01→10 is shown as an example of ND but patterns 11→10 and 01→11 are also instances of ND.

**Figure 3 entropy-22-00385-f003:**
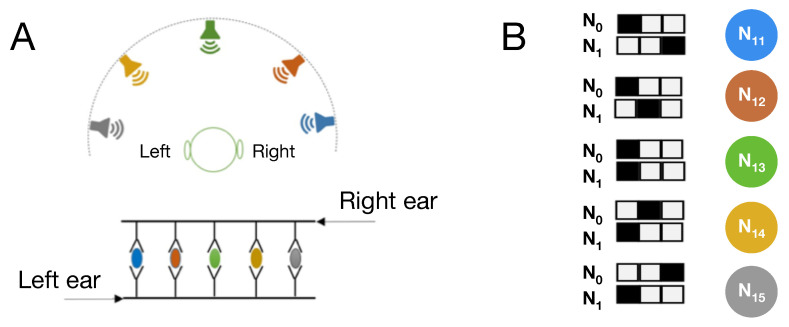
(**A**) Schematic of 5 sound sources at different angles with respect to a listener (top view) and Jeffress model of sound localization. (**B**) Schematic examples of 5 time sequences of input patterns received by the two sensory neurons (receptors of two ears) at three consecutive time steps. Black squares show presence of the stimuli in those neurons.

**Figure 4 entropy-22-00385-f004:**
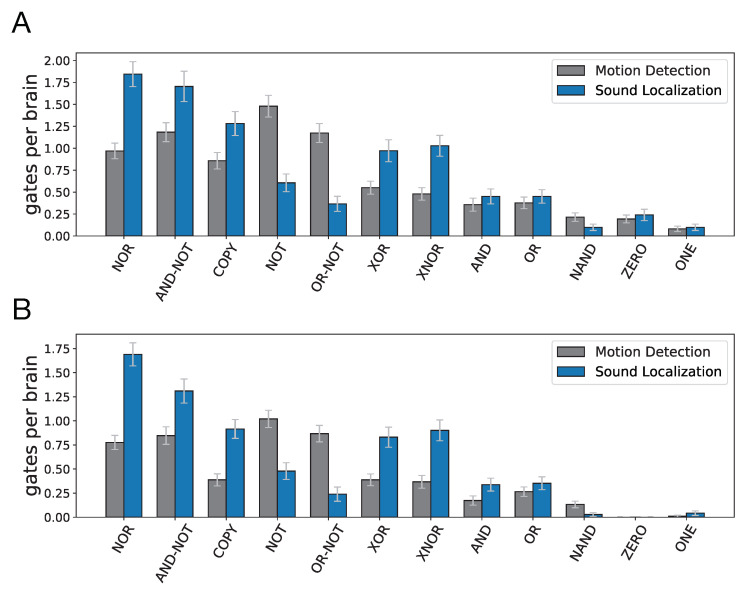
Frequency distribution of all, as well as essential, gates in evolved Markov Brains that perform the motion detection or sound localization task perfectly. (**A**) All gates. (**B**) Essential gates.

**Figure 5 entropy-22-00385-f005:**
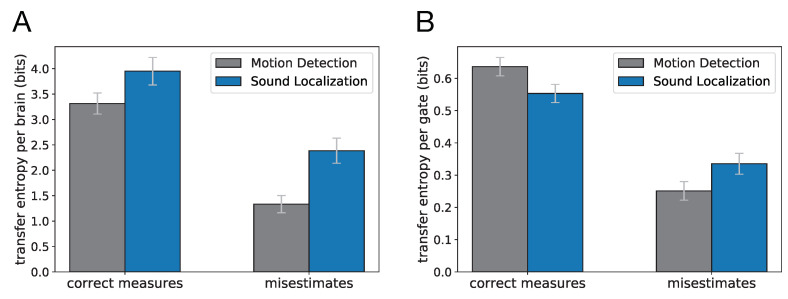
Transfer entropy measures, exact measures and misestimates by transfer entropy, on essential gates of perfect circuits for motion detection, and sound localization task. Columns show mean values and 95% confidence interval of misestimates and exact measures (**A**) per Brain, and (**B**) per gate.

**Figure 6 entropy-22-00385-f006:**
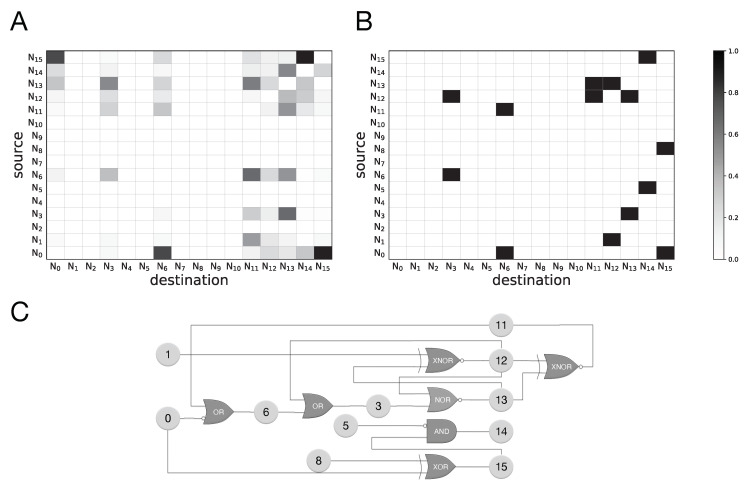
(**A**) Transfer entropy measures from neural recordings of a Markov Brain evolved for sound localization. (**B**) Influence map (also receptive field) of neurons derived from a combination of the logic gates connections and the Boolean logic functions for the same evolved Markov Brain, shown in (**C**). (**C**) The logic circuit of the same evolved Markov Brain; neurons $N_{0}$ and $N_{1}$ are sensory neurons, and neurons $N_{11}-N_{15}$ are actuator (or decision) neurons.

**Figure 7 entropy-22-00385-f007:**
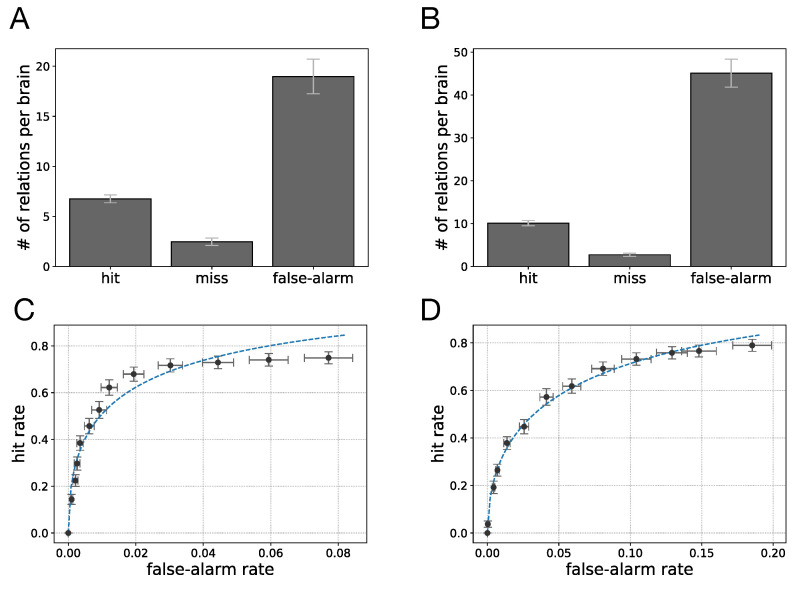
Transfer entropy performance in detecting relations among neurons of evolved (**A**) motion detection circuits, (**B**) sound localization circuits. Presented values are averaged across best performing Brains along with 95% confidence intervals. Receiver operating characteristic (ROC) curve representing TE performance with different thresholds to detect neurons relations in evolved (**C**) motion detection, (**D**) sound localization circuits.

**Table 1 entropy-22-00385-t001:** Transfer entropies and information in all possible 2-to-1 binary logic gates with or without feedback. The logic of the gate is determined by the value $Z_{t+1}$ (second column) as a function of the input $X_{t}Y_{t}$=(00,01,10,11). $H(Z_{t+1})$ is the Shannon entropy of the output assuming equal probability inputs, $TE_{X\to Z}$ is the transfer entropy from *X* to *Z*. In 2-to-1 gates without feedback, transfer entropies ${\mathrm{TE}}_{X\to Z}$ and ${\mathrm{TE}}_{Y\to Z}$ reduce to $I(X_{t}\;:\;Z_{t+1})$, and $I(Y_{t}\;:\;Z_{t+1})$, respectively. Similarly, transfer entropy of a process to itself is simply $I(Z_{t}\;:\;Z_{t+1})$ which is the information processed by *Z*.

			2-to-1 Network, Z=f(X,Y)			2-to-1 Feedback Loop, Z=f(Y,Z)		
Gate	$Z_{t+1}$	**$H(Z_{t+1}$)**	${\mathrm{TE}}_{X\to Z}$	${\mathrm{TE}}_{Y\to Z}$	TE Error	${\mathrm{TE}}_{Y\to Z}$	$I(Z_{t}:Z_{t+1})$	TE Error
ZERO	(0,0,0,0)	0.0	0.0	0.0	0.0	0.0	0.0	0.0
AND	(0,0,0,1)	0.81	0.31	0.31	0.19	0.5	0.31	0.19
AND-NOT	(0,0,1,0)	0.81	0.31	0.31	0.19	0.5	0.31	0.19
AND-NOT	(0,1,0,0)	0.81	0.31	0.31	0.19	0.5	0.31	0.19
NOR	(1,0,0,0)	0.81	0.31	0.31	0.19	0.5	0.31	0.19
COPY	(0,0,1,1)	1.0	1.0	0.0	0.0	1.0	0.0	0.0
COPY	(0,1,0,1)	1.0	0.0	1.0	0.0	0.0	1.0	0.0
XOR	(0,1,1,0)	1.0	0.0	0.0	1.0	1.0	0.0	1.0
XNOR	(1,0,0,1)	1.0	0.0	0.0	1.0	1.0	0.0	1.0
NOT	(1,0,1,0)	1.0	0.0	1.0	0.0	0.0	1.0	0.0
NOT	(1,1,0,0)	1.0	1.0	0.0	0.0	1.0	0.0	0.0
OR	(0,1,1,1)	0.81	0.31	0.31	0.19	0.5	0.31	0.19
OR-NOT	(1,0,1,1)	0.81	0.31	0.31	0.19	0.5	0.31	0.19
OR-NOT	(1,1,0,1)	0.81	0.31	0.31	0.19	0.5	0.31	0.19
NAND	(1,1,1,0)	0.81	0.31	0.31	0.19	0.5	0.31	0.19
ONE	(1,1,1,1)	0.0	0.0	0.0	0.0	0.0	0.0	0.0

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
