# Peer review of "Can Transfer Entropy Infer Information Flow in Neuronal Circuits for Cognitive Processing?"

_entropy, 2020, doi:10.3390/e22040385_

Round 1

Reviewer 1 Report

This article seeks to test the performance of the pairwise transfer entropy measure in inferring causal relationships, using standard logic gates and evolved Markov Brains as test beds. The authors emphasise whether the sum of detected pairwise transfers add up to the total information flowing from the sources to the target, including regarding whether what are called encrypted / obfuscated interactions are present. In testing, the authors also investigate the extent to which false positives are inferred as well.

These are interesting questions of course, but the two major issues I have with this article is that in attempting to address its goal - "Can Transfer Entropy Infer Information Flow in Neuronal Circuits for Cognitive Processing?" - it is: i. seemingly unaware of best practice in how the transfer entropy is used and interpreted in this setting; and ii. also uses the transfer entropy in rather inappropriate ways when it is used.
For these reasons, I cannot recommend acceptance.

First, the authors' main contribution is a demonstration that in evolved brains the pairwise transfer entropy can infer false links and miss some links. This is very well established in the neuroscience literature in general, and the authors seem unaware of the best practices around how transfer entropy is used and interpreted in this setting, in particular regarding its use in combination with conditional / higher order transfer entropies.
To start, it is well known that transfer entropy and causality (in the Pearl sense) are different concepts - long before the papers of James and Crutchfield built on here, the definitive series of papers providing analysis of the meaning and how the two are related here were [R1-3, below]. With that said, despite transfer entropy not being a causal measure, there are good reasons to try to use it to reveal causal structure from time series observations where interventionist measures are not appropriate, as the authors do here.
There is a rich literature of how this is attempted, particularly in neuroscience, some of which the authors do refer to at line 44. What is missed here is how well-known it is that pairwise transfer entropy is inadequate, and that (to some degree) conditional / higher order transfer entropies can address these issues. E.g., false positives due to sources duplicating information about the target can be removed by conditioning, and encrypted / obfuscated interactions (usually described as synergies) can be detected. A starting point for such literature would be the review in [R4, sec 7.2] and references therein. Crucially, consideration of conditional / higher order transfer entropies explains why the pairwise transfer entropies alone do not sum to give the total information at the destination variable (whereas a sum of iteratively conditioned transfer entropies via a properly formulated chain rule do [R4, sec 4.2.3]), and they should not be expected to do so. The authors seem unaware of this chain rule relation involving conditional transfer entropy terms - e.g. lines 83-86 seem genuinely surprised, and lines 94-97 rely on calculations for the single source case to prove this relation rather than the obvious mathematical formula. As such, the authors never describe why using pairwise transfer entropies alone would be inadequate for accounting for all flow to the target, nor consider how to address that. The closest the text comes to this is flirting with the partial information decomposition at line 67, which may satisy a desire to identify unique source contributions but (at this point) is not known to benefit network inference which is the main concern here.
The contemporary approaches to using transfer entropy for network inference referred to above (plus more recent contributions which validate the approach on large networks [R5-6]) also propose algorithms for how to combine the pairwise and higher order transfer entropies, and importantly seek to infer *sets* of sources as parents (see e.g. discussion in [R6]). The latter view of source sets contrasts with the obviously problematic, mechanistic interpretation at lines 97-101 which want to attribute information flows individually and uniquely to specific sources rather than collectively.
In light of the above, your article needs to rethink what it is adding in the context of the existing literature. That pairwise TE alone is inadequate for inference is clearly not new, and to a large degree dealt with. How those inadequacies are revealed in different circumstances may be new, if you could characterise the transfer entropies well.

This raises my second major point though, that transfer entropy is not appropriately measured here in the experiments, as follows.

Section 3.2 - my understanding here is that you have grabbed the TE values from the logic gates in table 1 in order to generate these results. Table 1 assumes independent source inputs, and no temporal correlations, and those assumptions follow through then to here. Those are strong assumptions, which you simply won't see in naturally embedded systems. Certainly the sources are not independent - see e.g. the feedback loop joining 12 and 13 in fig. 6. Temporal correlations naturally fall from how the input signals in motion detection and sound localisation are structured, and also the bias in the output of one gate (even if the inputs are unbiaed) leads to correlations which then feed into another gate. Those alone are enough to bring the results into question. Perhaps more important though is the claim that the results here are a "lower bound" on the error. This is stated as an apparent fact, without any serious argument or proof. I can't see any reason for that. If I reduce the entropy on the inputs for example such that the output entropy is reduced as well, I can arbirtrarily reduce the "misestimate" - so how is what you measure a lower bound? What are we to make of these results then?

Section 3.3 - there are simply not enough details here to repeat the experiment on how transfer entropy was calculated. How many time steps are you using? How were the inputs driven? The text simply says that you "collect recordings in all possible trials for each cognitive task" - what does that mean? You supply each distinct input once only? Do you supply them once, then wait a certain number of time steps for them to flow through the network? Do you change the inputs every time step or so before the previous has flowed all of the way through? This is completely unclear, and has significant ramifications for the interpretation.

Following on from the above discussion regarding 3.2, my impression is that temporal structure of time series are not being considered in this paper - e.g. with input configurations fed independently for raw logic gates, use of k,l=1, and my best guess at how inputs are treated in sec 3.3. Yet intracacies of temporal structure are crucial, and consideration of them are what distinguish transfer entropy from mutual information for example; to ignore them misunderstands the role of transfer entropy. E.g. table I and therefore section 3.2 are just mutual information results, since there is no temporal structure. It's not clear whether any temporal structure enters in section 3.3, since as above I can't tell how that experiment was run. If you're not interested in temporal structure, why investigate transfer entropy when you could just look at mutual information? One wonders whether it is simply to jump on the James and Crutchfield bandwagon regarding transfer entropy.
Similarly, the use of k=l=1 also implies that temporal structure is not considered important here. The authors claim that "the optimal values are obviously k=l=1" since this is a 1st order Markov process, however - to start with - this misses the fact that there are multiple sources involved in the interaction and only one is examined by the pairwise TE. Longer values of k can make a significant difference in this situation and otherwise (see e.g. use of k up to 16 for cellular automata, which are also 1st order Markov, in [R7]). Indeed, in the tasks selected here which are likely to have temporal structure, that could be very important.
Further, the inadequate selection of k=1 here is likely to have contributed to the "causally impossible" backwards flows noted at line 383 (and possibly others). For example, neuron 15 (a child of node 0) can add information regarding nueron 0's next step, given neuron 0's previous step, if there is information further back in the past of neuron 0 than the previous step (which would have then flowed to neuron 15 and been detected by TE). This could be the case where the inputs have temporal structure. It is well-known that inadequate k values can lead to false positives - this backwards situation is precisely described in your ref [14] - again standard practice which the authors do not seem aware of.

The ground truth influence map in Fig 6B also makes little sense in comparison to the circuit in 6C. How are 14 and 15 in the receptive field of 3, they are in completely different subcircuits? How is 14 in that of 6? How is 0 influenced by 8 and 15? Something is not right here.

Minor:
- equation (1) - I would suggest adjusting notation such that k,l >= 1 (in line with common use and what I think is your interpretation at line 370) rather than k,l >= 0 here.
- lines 155+ after Fig 2 - it might be worth clarifying that a "-1" result from the circuit is mapped to a 0, and that the 3 input patterns mapping to PD are (10->01,11->01,10->11).
- line 196 - how precisely do you knock out a gate - do you select one input at random and hard wire it through to the output? Or do you remove the gate entirely from the circuit and change its output wire to be fed from a constant 0 or 1? Or something else?
- line 271-2 - how does more than one logic gate write to a neuron, and how do you handle this situation?

References:
[R1] Ay and Polani, Advances in Complex Systems 11(17), pp. 17-41 (2008)
[R2] Lizier and Prokopenko, "Differentiating information transfer and causal effect", European Physical Journal B 73(605), 605-615, 2010
[R3] Chicharro and Ledberg, "When Two Become One: The Limits of Causality Analysis of Brain Dynamics", PLoS ONE, 7(3), e32466, 2012
[R4] Bossomaier et al., "Introduction to transfer entropy", Springer, 2016
[R5] Sun et al., "Causal Network Inference by Optimal Causation Entropy", SIAM Journal on Applied Dynamical Systems, 14(1), 73–106, 2015
[R6] Novelli et al., "Large-scale directed network inference with multivariate transfer entropy and hierarchical statistical testing", Network Neuroscience, 3(3), 827–847, 2019
[R7] Lizier et al., "Local information transfer as a spatiotemporal filter for complex systems" Physical Review E, 77(2), 026110, 2008

Reviewer 2 Report

The idea presented in the paper (to use transfer entropy notion to infer causal relations between random  variables) is interesting, but the design of the numerical experiments presented in the paper is so complex that it is almost impossible to answer precisely to the question addressed at the beginning. The final message contained in the paper, stripped to the bones, is : in some cases it works reasonably well, in others not, so be cautious in using this transfer entropy tool.

Another concern is in the choice of Markov process to design the evolving network of logic gates. Since these processes are memory less, the  authors do not make full use of the strength of the transfer entropy notion, which can take into account an arbitrarily long  record of the past  of the two random variables.

To improve the readability of the paper, I suggest to first discuss the simplest conceivable example of network with the highest possible accuracy and then move to more ambitious realm of evolving digital brains. I think that the average reader of Entropy would benefit from this.

Reviewer 3 Report

In the manuscript, the authors add computational models of cognitive agents as a way to evaluate the degree to which transfer entropy sometimes inadequately estimate casual interactions. In the first part of the paper, the authors analyze transfer entropy in all possible 2-to-1 binary logic gates with and without feedback. In the second part of the paper, the authors use artificial evolution to train networks of binary gates (Markov Brains) to perform two different tasks (a motion detection task and a sound localization task), the authors analyze the gate composition in the resulting solutions, and then also how well transfer entropy estimates the flow of information in those circuits. Their results highlight the limitations of transfer entropy, particularly as applied to systems that perform different tasks. 

The paper is well written and the results are clearly presented. I only have a few minor suggestions for improvement. 

The caption for Figure 1B is incorrect. And more importantly, how the feedback works in the 2-to-1 binary logic with feedback is not clearly explained.  It is surprising that both tasks created circuits with such a similar profile of logic gates. Perhaps it would be useful to analyze a circuit with an entirely different profile.  Finally, the authors do not go in any depth into an explanation for whether there is a relationship between the composition of the gates and the misestimation in the information flow by transfer entropy. Can a formal relationship be established or derived from the analysis? 

Round 2

Reviewer 1 Report

I thank the authors for the significant efforts they have made to address my comments.

Several points remain:

1. The authors have responded to my primary comment as summarised in their statement: "This comment reminded us to emphasize to the reader that we are NOT using transfer entropy to infer causality, but rather to detect and follow the flow of information through the neurons of the brain over time".
Yet it is still difficult for the reader to reconcile that explanation with what is in the text. The abstract is completely focussed on inferring causality (with "causal" coming up 6 times there I think) -- e.g. "whether ... can infer information flow using the transfer entropy concept, when compared to a ground-truth model of causal influence constructed from connectivity and circuit logic". That sounds a lot like a focus on causality rather than the "flow of information".
The explanations inserted into the Introduction don't add to my understanding of what you mean. The first half on the subtleties of using TE to infer causality is a good addition, but the only part that seeks to make the distinction above is: "In some sense, inferring information flow is a much simpler task than finding all causal influences, as we need only to follow direct causal influences" - and I can't decipher the distinction you're trying to make from that.
With all that said, your meaning as I understand it is of whether all of the information is accounted for by pairwise TEs alone as compared to H(Z_t) in (5) and others (which have been inserted in response to review 1, which is good). I think you need to find a more clear way to articulate that as the meaning, because the phrasing in the abstract and earlier parts of the introduction are not conveying that.

Similarly, whilst you have inserted the chain rule combining pairwise and higher order transfer entropies in (2), the later text continues to dismiss not only pairwise TE but also conditional TE because neither alone can accurately account for the full H(Z_t). The smart reader can glean from (2) that both forms are required to properly account for the flow (in your meaning as well as mine), but you never point this out to the reader. As per my first review, this is an important point which is well recognised in the neuroscience literature. Without noting this, your text simply misses the point that if you want all information flow accounted for, you have to include higher order transfer entropies in conjunction with pairwise as per that chain rule. Your goal of showing when and where the pairwise alone is not adequate to account for the whole flow is fine, but without pointing out that the only thing that always works is combining both pairwise and conditional forms, you seem to be obfuscating what is going on.

2. Along similar lines, the additional math inserted in the introduction is certainly a good idea, but serious problems remain.
The main issues centre on the multivariate terms like I(Y_t : Z_t : Z_{t+1} ) inserted into Fig 1. After (4) the conditional variant is defined as a conditional MI. It absolutely is not that - whilst in some sense it may be considered an extension of that, it is a (conditional) co-information term. The mislabelling of that term is highly, highly misleading. Indeed, the use of a Venn diagram in Fig 1C/D suggest these terms are positive - this is well known not to be the case, as shown in one of your examples and of course recently explored in the partial information decomposition literature. Using such terms at all is highly confusing for the uninitiated reader. If you want to use such Venn diagrams with information terms, the partial information decomposition is the only rigorous method - these approaches are seriously outdated and misleading.
Further - Eqn (7) is just wrong. H(Z_{t+1}) = T_{Y->Z} + I(Z_t;Z_{t+1}) + H(Z_{t+1} | Z_t, Y_t). And H(Z_{t+1} | Z_t, Y_t) (equal to 0 in this situation) is not equal to I(Z_{t+1} : Z_t : Y_t) either in general or in this case. Where did this come from? Does this lead to other errors elsewhere?

3. There's no response to a large block of my comments starting from "Following on from the above discussion regarding 3.2..." on p. 4. None of the serious questions I raised with the results of sec 3.2 there are addressed. This seems a rather large oversight.
For example, the comments on the inadequate selection of k=1 towards the bottom seem completely ignored. Unless the new text in the Discussion was meant to address this (?); that made some claims that you could restrict yourself to k=1 which is demonstrably not correct (as per my original comments, just because the system as a whole is first-order Markov one can demonstrate false positives if this is used on any single pair).

Author Response

See pdf file. 

Reviewer 2 Report

I have no comments nor suggestions

Author Response

We thank the reviewer for their time.